

# Impact of inter- and intra-individual variation, sample storage and sampling fraction on human stool microbial community profiles

Yun Kit Yeoh[1,2], Zigui Chen[1,2], Mamie Hui[1,2], Martin C.S. Wong[2,3], Wendy C.S. Ho[1], Miu Ling Chin[1], Siew C. Ng[2,4,5], Francis K.L. Chan[2,4] and Paul K.S. Chan[1,2]

[1] Department of Microbiology, Faculty of Medicine, The Chinese University of Hong Kong, Shatin, Hong Kong SAR, China
[2] Centre for Gut Microbiota Research, Faculty of Medicine, The Chinese University of Hong Kong, Shatin, Hong Kong SAR, China
[3] Jockey Club School of Public Health and Primary Care, Faculty of Medicine, The Chinese University of Hong Kong, Shatin, Hong Kong SAR, China
[4] Department of Medicine and Therapeutics, Faculty of Medicine, The Chinese University of Hong Kong, Shatin, Hong Kong SAR, China
[5] State Key Laboratory of Digestive Disease, Li Ka Shing Institute of Health Sciences, The Chinese University of Hong Kong, Shatin, Hong Kong SAR, China

Corresponding author
Paul K.S. Chan,
paulkschan@cuhk.edu.hk

## ABSTRACT

Stools are commonly used as proxies for studying human gut microbial communities as sample collection is straightforward, cheap and non-invasive. In large-scale human population surveys, however, sample integrity becomes an issue as it is not logistically feasible for researchers to personally collect stools from every participant. Instead, participants are usually given guidelines on sample packaging and storage, and asked to deliver their stools to a centralised facility. Here, we tested a number of delivery conditions (temperature, duration and addition of preservative medium) and assessed their effects on stool microbial community composition using 16S rRNA gene amplicon sequencing. The largest source of variability in stool community composition was attributable to inter-individual differences regardless of delivery condition. Although the relative effect of delivery condition on community composition was small compared to inter-individual variability (1.6% vs. 60.5%, permutational multivariate analysis of variance [PERMANOVA]) and temporal variation within subjects over 10 weeks (5.2%), shifts in microbial taxa associated with delivery conditions were non-systematic and subject-specific. These findings indicated that it is not possible to model or accurately predict shifts in stool community composition associated with sampling logistics. Based on our findings, we recommend delivery of fresh, preservative-free stool samples to laboratories within 2 hr either at ambient or chilled temperatures to minimise perturbations to microbial community composition. In addition, subsamples from different fractions of the same stool displayed a small (3.3% vs. 72.6% inter-individual variation, PERMANOVA) but significant effect on community composition. Collection of larger sample volumes for homogenisation is recommended.

## INTRODUCTION

The human gut microbiome has gained unprecedented appreciation for its many vital roles in health and disease. In recent years, studies have linked dysbiosis in gut microbial communities to conditions such as obesity (*Turnbaugh et al., 2006*), diabetes (*Qin et al., 2012*) and even neurodegenerative diseases such as Parkinson's (*Bedarf et al., 2017*) and Alzheimer's disease (*Vogt et al., 2017*), underscoring their roles in development, maturation and maintenance of health. Owing to this intimate association between gut microorganisms and their hosts, the gut microbiome has become an attractive target for healthcare practices that aim to restore and promote health by manipulating microbial communities of the gut. This undertaking requires an underlying knowledge of the "healthy" baseline composition and function of gut microbial communities, and how they respond to host factors and environmental perturbations. To that end, various studies have examined the influence of demographics, environment and lifestyle habits over ecology of gut microbial communities in cohorts from various biogeographical backgrounds (*Zeevi et al., 2015*; *Zhernakova et al., 2016*; *Falony et al., 2016*; *Rothschild et al., 2018*), and generally agree that environment is a stronger determinant of gut microbiota composition relative to host factors such as ancestry and genetics.

Currently, there are no reference gut microbiota profiles for the South Asian population in Hong Kong. Since environment is commonly the strongest predictor of gut microbiota profiles (*Rothschild et al., 2018*), it is necessary to assess whether the gut microbiota of distinct populations are comparable to previously surveyed cohorts and validate whether trends observed from international studies apply to a local population while considering differences in demographics and lifestyles. To address this lack of a local gut microbiome dataset, we are initiating a gut microbial community survey targeted at the general Hong Kong public. One important consideration is that gut microbiome surveys face a challenge of balancing sampling logistics and study power (*Sze & Schloss, 2016*) where hundreds to thousands of stool samples are often needed to account for unique population niches and discern potential confounding factors. Because of the large number of participants a population gut microbiome survey would entail, we wanted to formulate simple instructions for participants to deliver stools to a centralised laboratory while also preserving sample integrity as much as possible during transit.

Here, we tested the impact of commonly used sample delivery conditions on stool microbial community profiles. Parameters evaluated included delivery temperature, duration and use of a preservative medium. The impact on microbial community composition was assessed by 16S ribosomal RNA gene (16S) amplicon profiling, and results were interpreted to determine the relative contributions of the evaluated parameters towards compositional differences in community composition. In addition, shifts in the relative abundances of specific microbial taxa associated with the tested parameters were assessed to determine whether perturbations in community composition due to sampling logistics could be generalised across samples and predicted in future sampling efforts.

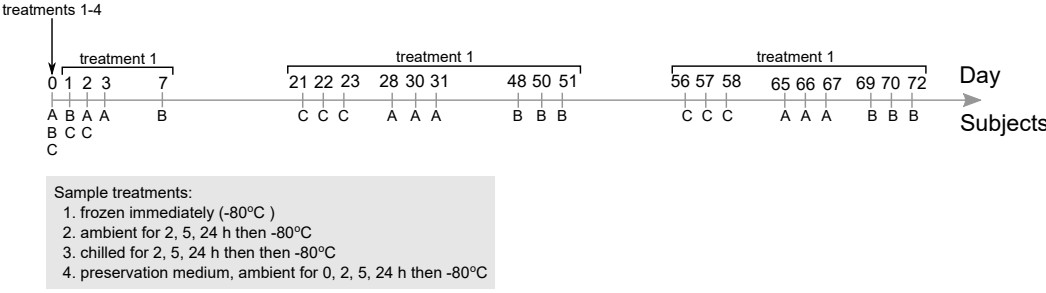

**Figure 1** Overview of the stool microbial community survey and main experimental factors investigated. Three subjects were recruited to provide stools three times over the span of three to four days (with the exception of subject B at day 0, 1 and 7) thrice over the duration of 72 days. Stools collected at day 0 were subjected to all four treatments to assess their influence on microbial community composition. Community profiles from stools immediately frozen at −80 °C used as references for comparison. Subsequent stools collected from day 1 onwards (sampling time points indicated in figure) were immediately frozen at −80 °C only treatment 1).

# MATERIAL AND METHODS

## Ethics statement and consent

This study has been approved by the Joint Chinese University of Hong Kong-New Territories East Cluster Clinical Research Ethics Committee (reference number 2016.707). Written consent was obtained from all participants prior to collecting stool samples.

## Sample collection

Stool samples were collected from three subjects on nine separate occasions. Collection was scheduled in three batches—stools were collected thrice over a period up to seven days to study short-term temporal variation in stool microbial communities, and again three to four weeks and eight to ten weeks after the first collection to study medium-term temporal variability (Fig. 1). Subjects defecated on site and the whole stool was collected in a sterile container. The first stool collected from all three subjects at day 0 were aliquoted and assigned to four treatments intended to simulate how various delivery conditions could impact outcomes of 16S-based gut microbial community surveys: (i) frozen at −80 °C within 5 min of defecation, (ii) ambient temperature for 2, 5 and 24 h then frozen at −80 °C, (iii) chilled with cool packs for 2, 5 and 24 h in an insulated container then frozen at −80 °C, and (iv) stored in a stool nucleic acid collection and preservation medium (catalogue number 63700; Norgen Biotek Corp., Canada) and frozen at −80 °C immediately or after 2, 5 and 24 h at ambient temperature. Subsequent stool samples collected from day 1 onwards were immediately frozen at −80 °C.

## DNA extraction and 16S amplicon sequencing

DNA was extracted in triplicate from 0.1 g homogenised fractions of stool using the QIAGEN DNeasy PowerSoil Kit following manufacturer's instructions. Following extraction, the V3 and V4 variable regions of the 16S gene were amplified by polymerase chain reaction (PCR) in 25 μl reactions with the following master mix recipe: 1X

PCR buffer, 0.2 mM of each dNTP, 1 mM MgCl$_2$, 0.2 $\mu$M of each forward (341F: CCTACGGGNGGCWGCAG) and reverse (806R: GGACTACNVGGGTWTCTAAT) primers (*Klindworth et al., 2013*; *Apprill et al., 2015*), 0.625 U Hot Start plus DNA Taq polymerase and molecular biology grade water. Thermocycler settings were as follows: 95 °C for 3 min, followed by 15 cycles of 94 °C for 45 s, 55 °C for 30 s and 72 °C for 3 min, followed by 10 cycles of 94 °C for 45 s, 60 °C for 30 s and 72 °C for 3 min, and finally 72 °C for 10 min. Four DNA-free negative controls using water were included in the PCR. Successful PCR amplification was verified by visualising products on an agarose gel. Sequencing adapters and multiplex indices were added to the PCR products in a second 25-$\mu$l PCR reaction with the following master mix recipe: 1X PCR buffer, 0.2 mM of each dNTP, 1 mM MgCl$_2$, 0.2 mM of each Nextera index 1 and 2 primers, 0.1 $\mu$M of each forward and reverse Golay index, 0.625 U Hot Start plus DNA Taq polymerase and molecular biology grade water. Thermocycler settings were as follows: 95 °C for 3 min, followed by 10 cycles of 94 °C for 45 s, 63 °C for 30 s and 72 °C for 3 min, and finally 72 °C for 10 min. Amplicons were purified using the QIAquick Gel Extraction Kit. Purified final products including the four negative controls were sent to the Genomics Resource Core Facility at Cornell University for 2 × 300 bp sequencing on an Illumina MiSeq using the v3 MiSeq Reagent Kit.

## 16S sequence data processing

Demultiplexed raw sequence data were imported into QIIME 2 v2017.12 (https://qiime2.org/). Using the DADA2 workflow (*Callahan et al., 2016*) in QIIME 2, primer and low quality sequences were trimmed, and remaining reads subsequently denoised and merged. Alpha diversity metrics were calculated based on the counts table produced by DADA2, normalised to 1,086 sequences per sample (Fig. S1D). To assign taxonomy to the sequences, a classifier was first trained on sequences extracted from the SILVA 16S database release 128 (*Quast et al., 2013*) using the 16S gene V3-4 universal primer sequences. This classifier was then run on the representative sequences produced by DADA2 to assign probable taxonomies to the corresponding sequences. The final counts table based on exact sequence variants (ESVs) was exported from QIIME2, chloroplast sequences removed, and used as input in R for statistical analysis.

## 16S amplicon-based microbial community composition analyses

The resulting ESV counts table from QIIME2 was imported together with sample metadata into R v3.4.1. A centered log-ratio transformation was applied to ESV counts before downstream analyses to ensure that the counts fulfilled assumptions of independence between predictor variables for statistical analyses (explained in *Lê Cao et al., 2016*). Permutational multivariate analysis of variance (PERMANOVA) was used to assess whether factors such as subject-specificity, sampling time point and the simulated storage conditions significantly influenced community composition, as well as to determine the amount of variation in community composition attributable to each of these experimental factors. Principal component analysis (PCA) was used to visualise the clustering of samples based on their compositional similarities. Association of ESVs to experimental factors

were tested using generalised linear models (GLMs) and sparse partial least squares discriminant analysis (SPLSDA). PERMANOVA, PCA and GLMs are implemented in the vegan R package v2.4-6 (*Oksanen et al., 2013*, and SPLSDA implemented in the mixOmics R package v6.3.1 (*Rohart et al., 2017*). Statistical power was assessed using the HMP R package (*La Rosa et al., 2012*). Figures were edited in Inkscape v0.92 for clarity. All data are provided as supplemental files 1 to 4.

## Data Availability

Raw sequence data generated for this study are available in the Sequence Read Archive under BioProject accession PRJNA450690.

## RESULTS

### Effect of storage conditions relative to inter-individual differences and intra-individual temporal variation

PCR amplicons from a total of 189 stool samples (Table S1) were pair-end sequenced on an Illumina MiSeq platform producing 886,252 reads, with a median of 4,556 reads per sample. Reads were quality-filtered to remove adaptor, primer, low-quality and chloroplast sequences, producing a final count of 475,361 reads. Of four DNA-free negative controls included in the PCR and sequencing workflow, two contained zero reads while the other two had three and four reads each. These controls were excluded from downstream statistical analyses. A total of seven samples each containing less than 1,000 reads were also excluded, leaving 182 samples.

To determine which parameters were most strongly associated with community composition, we first applied a centered log-ratio transformation on 16S counts data and then performed a PCA. Samples clustered according to their respective subjects based on community composition regardless of storage conditions or sampling time point (Fig. 2D), indicating that inter-individual variation had an overarching influence when comparing stool microbial communities among multiple subjects. Subject-to-subject stool microbial community specificity was also partially reflected in alpha diversity metrics including observed species, Shannon and Faith's phylogenetic diversity indices, in which samples from one subject (subject B) were consistently more species rich and phylogenetically diverse compared to samples from the other two subjects across all nine sampling time points ($p < 0.001$, Kruskal–Wallis test) (Fig. S1). Next, we performed PCA on samples from each subject separately and observed secondary clustering by sampling time point and transit storage conditions in all three subjects (Figs. 2A–2C). Stool communities collected across multiple days tended to cluster together relative to samples from weeks apart, reflecting shifts in community composition over short and medium-term durations (Fig. S2). This overall sequence of sample clustering beginning with between-subject variability, followed by within-subject temporal variability and sample storage conditions was supported by a permutational multivariate analysis of variance (PERMANOVA) indicating that subjects (accounting for 60.5% of the total variance in community composition), sampling time point (5.2%) and storage conditions (1.6%) were significantly associated with community composition in descending size of

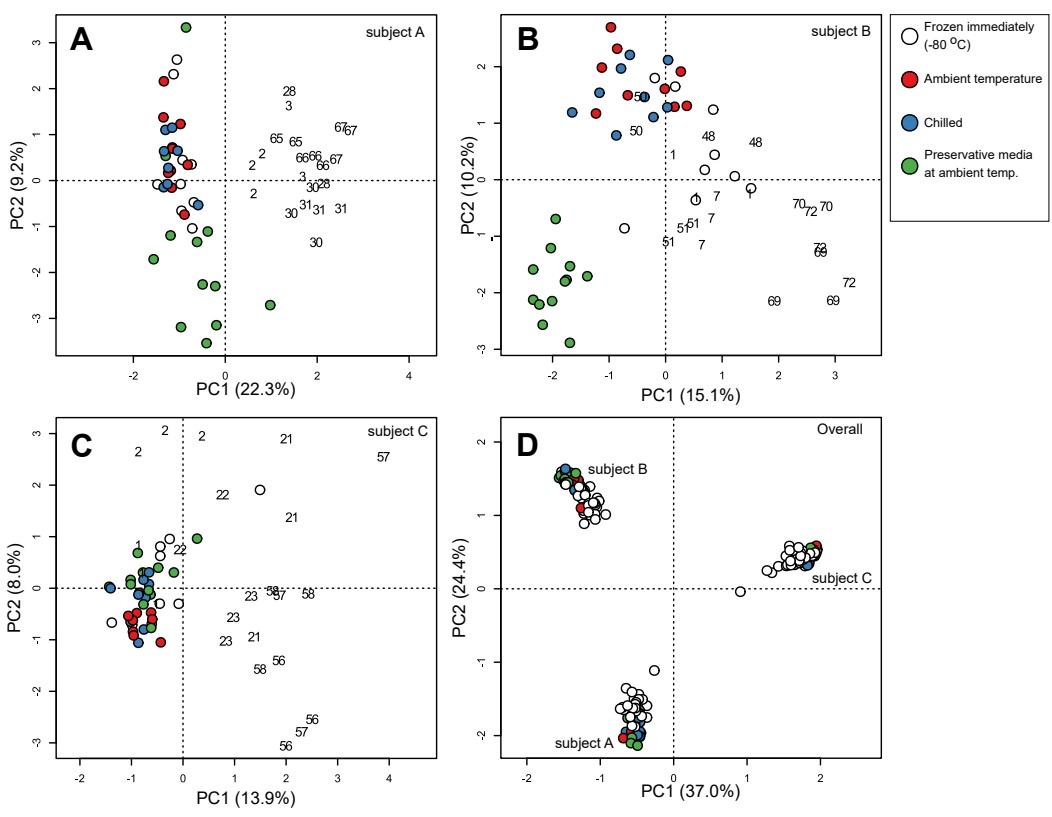

**Figure 2** **Principal component ordination of stool microbial community composition collected from three subjects over 72 days.** Community composition was determined by sequencing 16S rRNA gene amplicons. Ordination of (A) samples from subject A; (B) samples from subject B; (C) samples from subject C; (D) samples from all three subjects. Each circle represents one sample, and colour of circle represents storage condition: white, immediately frozen at −80 °C red, ambient temperature; blue, chilled; green, preservative medium at ambient temperature. Numbers in (A–C) represent number of days subsequent samples were collected following initial sampling at day 0 (day 0 samples immediately frozen are represented by the white circles). All samples collected after day 0 were immediately frozen at −80 °C.

effect (Table 1). Storage duration (2, 5 or 24 h) before transferring samples to −80 °C were also significantly associated with shifts in community compositions in the three storage conditions (ambient, chilled and preservative) (PERMANOVA, Table 1).

## Effect of preservation medium on stool community composition

To specifically investigate the effects of sample storage conditions during transport to the laboratory on community composition, we performed PCA on community composition of stools that were subjected to all three storage conditions (i.e., stools from the first sampling time point on day 0). Samples were analysed separately by subject to remove the large influence of inter-individual variability. In all three subjects, PCA ordination of stool community composition revealed that samples stored in preservative medium often clustered to the exclusion of fresh frozen, ambient temperature or chilled samples (Figs. 3A–3C). In comparison, community compositions of samples without preservation buffer were comparatively more similar to each other irrespective of storage condition

**Table 1 PERMANOVA of all samples.**

| | Degrees of freedom | Sums of squares | F model | $R^2$ | $P$ value |
|---|---|---|---|---|---|
| subject | 2 | 58,888 | 198.1 | 0.605 | <0.001 |
| sampling time point | 8 | 5088 | 4.3 | 0.052 | <0.001 |
| storage temperature | 3 | 1597 | 3.6 | 0.016 | <0.001 |
| subject:sampling time point | 16 | 9071 | 3.8 | 0.093 | <0.001 |
| storage temperature:duration | 7 | 1184 | 1.1 | 0.012 | 0.04 |
| Residuals | 145 | 21555 | | 0.221 | |

**Notes.**
    **subject**: three subjects A, B and C.
    **sampling time point**: days stools were collected.
    **storage temperature** before long-term freezing: immediately frozen, ambient, chilled, preservative at ambient.
    **duration** of storage before long-term freezing: 0, 2, 5 or 24 h.

(Fig. 3A–3C). Interestingly, effects of the preservation medium on alpha diversity was only significant in one subject (subject B, Kruskal–Wallis tests, $p < 0.05$) (Fig. S3), indicating that the preservation medium tested here introduced inconsistent biases into the resident stool community across samples. To examine which ESVs were enriched in the preserved samples, we used a GLM to predict ESV associations with the use of preservative medium based on centered log-ratio transformed 16S counts. Using data from all three subjects, the model predicted 51 ESVs significantly associated with preservation medium treatment (false discovery rate-adjusted $p < 0.05$), 44 of which were enriched in preserved samples (Table S2). These associations were supported by an SPLSDA implemented in the mixOmics R package, in which 37 ESVs were overlapping between the two methods (GLM and SPLSDA, Table S2). In total, 26 of the 37 ESVs were clostridial. When communities were analysed separately by subject, there was little overlap in the ESVs associated with preservation medium among subjects (Table S3) which was most likely due to the strong inter-individual variability in stool communities. Nevertheless, there was a strong representation of clostridial ESVs associated with the use of preservation medium in all three subjects (Table S3). These results suggest that the preservation medium tested here may be enriching for certain clostridial taxa in stools. Thus, its effects need to be further evaluated in other stool and non-stool samples to determine whether the observed enrichment is specific to this study. Since these findings indicated that community composition in samples treated with preservation medium was altered relative to preservative-free options, we revisited the initial analysis of all 182 samples and excluded those stored in preservation medium. When the PERMANOVA was restricted to non-preserved samples, sample storage (ambient vs. chilled temperatures) now accounted for a smaller but still significant amount of variability (0.6% vs 1.6% when preserved samples were included), and duration of storage was no longer associated with community composition when analysing stools from multiple subjects combined (Table 2).

## Effect of storage duration on community composition

To more specifically identify at which time points significant differences could be detected in the community composition of stools stored at ambient and chilled temperatures relative

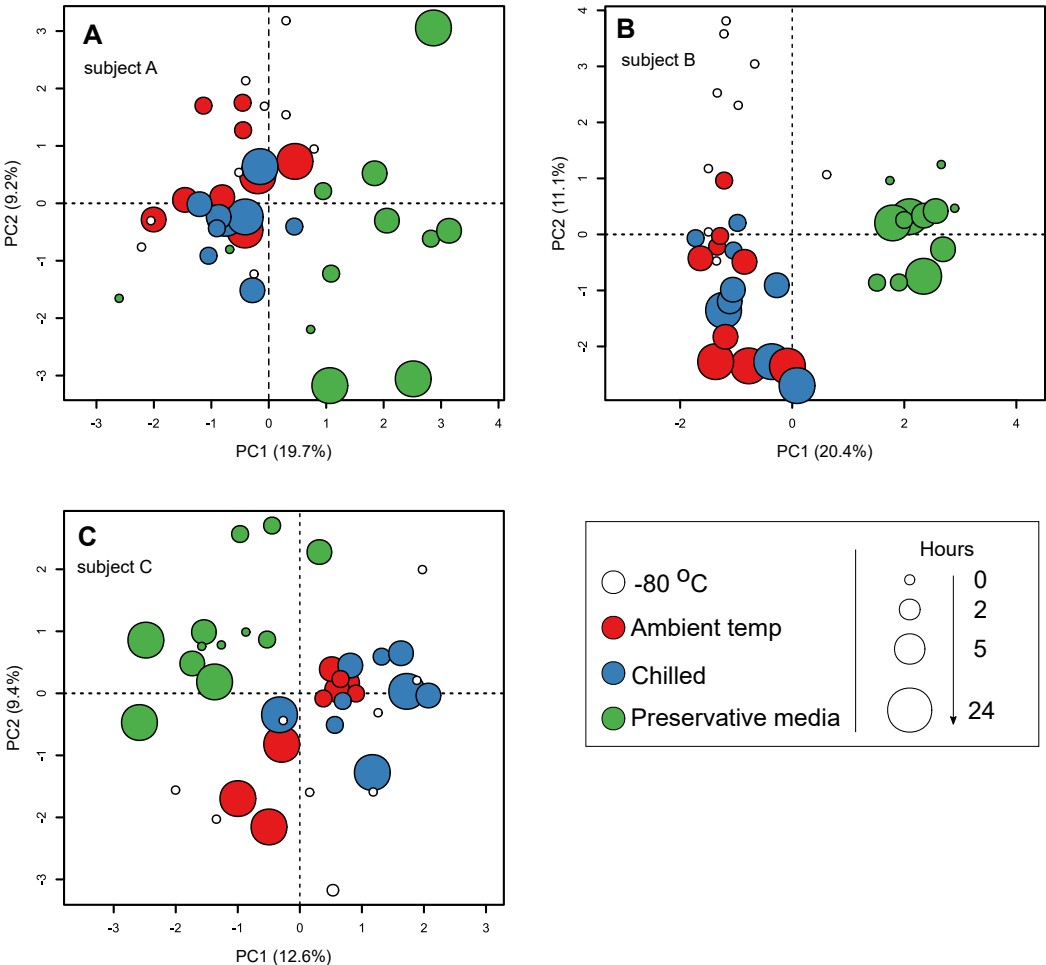

**Figure 3** **Principal component ordination of stool microbial community composition by storage condition and duration.** Community composition was determined by sequencing 16S rRNA gene amplicons. (A) samples from subject A; (B) samples from subject B; (C) samples from subject C. Each circle represents one sample. Colour of circle represents storage condition: white, immediately frozen at −80 °C red, ambient temperature; blue, chilled; green, preservative medium at ambient temperature. Size of circle represents duration samples were subject to their respective conditions before frozen long-term at −80 °C.

to fresh frozen samples, we separately compared profiles of samples from 2, 5, and 24 h to the 0 h reference using PERMANOVA. At 2 h, stools from all three subjects did not differ in community composition from their respective fresh frozen fractions regardless of ambient or chilled storage. At 5 and 24 h, only stools from subject B showed significant alterations in both ambient and chilled conditions whereas stool profiles of the other two subject did not differ from their respective 0 h fractions (Table S4). Similar to the non-systematic effects of the tested preservation medium on community composition, these observations further indicate that storage conditions can produce varying outcomes on stool community composition among individuals. We then assessed changes in the centered log-ratio transformed counts of each ESV across the 0, 2, 5 and 24 h time points using GLMs to
**Table 2   PERMANOVA excluding preserved samples.**

|  | Degrees of freedom | Sums of squares | F model | $R^2$ | P value |
|---|---|---|---|---|---|
| subject | 2 | 45,861 | 170.1 | 0.604 | <0.001 |
| sampling time point | 8 | 4895 | 4.5 | 0.064 | <0.001 |
| storage temperature | 2 | 476 | 1.8 | 0.006 | <0.001 |
| subject:sampling time point | 16 | 8923 | 4.1 | 0.117 | <0.001 |
| storage temperature:duration | 4 | 573 | 1.1 | 0.008 | 0.27 |
| residuals | 113 | 15237 |  | 0.2 |  |

**Notes.**
   **subject**: three subjects A, B and C.
   **sampling time point**: days stools were collected.
   **storage temperature** before long-term freezing: immediately frozen, ambient, chilled, preservative at ambient.
   **duration** of storage before long-term freezing: 0, 2, 5 or 24 h.

identify taxa enriched or depleted over time. A total of 86 ESVs were implicated, of which 74 were enriched, eight depleted, and four both enriched and depleted among the three subjects (Table S5). While the number of ESVs enriched or depleted could be influenced by the relative nature of microbial community data, their taxonomic identities were specific to subjects and included a range of common gut taxa (Table S5). These observations indicate that there is likely no generalizable pattern in how stool community composition from multiple individuals changes over a 24 h storage duration. Within subjects, many ESVs enriched or depleted over time in the ambient and chilled conditions overlapped, indicating that temperature may not radically influence enrichment/depletions at least in the first 24 h. Taken together, these results show that responses in stool community composition to storage conditions is specific to individuals and may not be easily accounted for in downstream analyses. While these shifts were small compared to inter-individual variability detected in stool community compositions (Table 2), their potential impacts on gut communities should be considered when interpreting survey results.

## Effect of sampling different stool fractions on community composition

For each of the three subjects, we produced replicate community profiles from three separate fractions of the whole stool (from day 0) taken at least 5 cm apart to assess variability in community composition attributable to heterogeneity in microbial niches within stools. Subsampling of different fractions was significantly associated with distinct community compositions in two of three subject ($p < 0.05$; PERMANOVA) (Table S6, Fig. S4). We then combined the samples from all three subjects and performed a PERMANOVA to assess the relative influences of inter-individual variability and intra-sample heterogeneity on stool community composition. The PERMANOVA indicated that in the absence of other experimental variables, differences in community composition linked to sampling different fractions of the same stools were subtle but still significant despite the stronger influence of inter-individual variation (72.6% vs. 3.3% $R^2$, PERMANOVA) (Table S6). This finding indicates that distinct fractions of the same stool may not necessarily be true biological replicates, therefore collecting a larger sample for homogenisation before storage

may be necessary. Taken together, the variables investigated in this study ranked according to their effect on stool microbial community composition are: inter-individual variation >intra-individual temporal variation >use of preservative media >storage temperature >storage duration (up to 24 h) before long term freezing.

## DISCUSSION

Large-scale population gut microbiota surveys often face a logistical issue of procuring hundreds of stools while maintaining sample integrity. Typically, the ideal laboratory procedure for storing stools is often immediate freezing at −80 °C following defecation (*Carroll et al., 2012*; *Fouhy et al., 2015*); however, this workflow is not feasible in surveys involving large numbers of participants from the general public. A major concern with collection of stool and other biological samples intended for microbial community surveys is the issue of prolonged exposure to ambient temperatures during transit, leading to shifts in community composition (*Cardona et al., 2012*; *Carroll et al., 2012*; *Guo et al., 2016*). Biological samples are usually chilled or mixed with preservative media to suppress opportunistic microorganisms from flourishing and dominating community profiles (*Vandeputte et al., 2017*), although this problem may be somewhat alleviated by setting a maximum allowed transit duration. A few studies have compared the use of preservation buffers in maintaining microbial community profiles and generally recommend its use especially when samples cannot be immediately frozen due to logistical constraints (*Menke et al., 2017*; *Flores et al., 2015*; *Voigt et al., 2015*). Our results, however, show that preservation media may not be necessary for human stool samples for short transit durations of up to 24 h as the use of a medium was associated with larger shifts in microbial community composition compared to samples stored at ambient and lowered temperatures (Fig. 3). While there is a possibility that the three samples collected in this study contained less appreciable abundances of fast-growing microorganisms favoured at ambient temperatures, our findings echo recommendations relating to storage options and durations made by Vandeputte and colleagues, in which they rated chilled buffer-free samples for up to 24 h as the best option other than immediate freezing (Table 1 in *Vandeputte et al., 2017*). Nevertheless, we only tested one preservative medium on stools from three subjects and there may be alternatives that produce community profiles more comparable to buffer-free storage conditions. Song and colleagues previously tested five sample preservative media on human stool samples and reported a general increase in Firmicutes taxa in samples preserved using ethanol (70% and 95%), RNALater or Whatman FTA cards but not OMNIgene Gut (*Song et al., 2016*). Our observations of increased relative abundances of clostridial ESVs (members of Firmicutes) in preserved samples (Table S2) was consistent with their findings. Although our data set presented here provided sufficient statistical power for discriminating preserved and non-preserved samples based on community profiles (98.5% power at 0.05 alpha level, Monte-Carlo permutation of Dirichlet multinomial likelihood ratio implemented in HMP R package), additional tests inclusive of a larger number of subjects are required to determine whether composition and/or diversity of the resident stool microbial community influence performance of preservative media.

As highlighted above, our findings indicated that preservative media may not be necessary for stools collected and frozen within 2 h (Fig. 3, Table S4). Stools stored at ambient and chilled temperatures produced community profiles more closely resembling those from fresh frozen samples, however, we noted that these outcomes were variable between subjects (Table S4). A few published studies have reported that storing stools at 4 °C for 24 h did not significantly shift community profiles compared to fresh frozen −80 °C controls (*Bassis et al., 2017*; *Choo, Leong & Rogers, 2015*; *Tedjo et al., 2015*; *Cardona et al., 2012*). While profiles from two of our three subjects did not significantly change over 24 h at ambient and chilled temperatures, significant shifts were detected in the stools of the third subject (subject B) at five hours post collection. Furthermore, when we attempted to identify taxa associated with these shifts, we observed that the relatively enriched and depleted ESVs were specific to subject and encompassed generic human gut taxa including various *Bacteroidales* and clostridial members (Table S5). In some instances, the same ESV present in multiple subjects showed enrichment in one but depletion or no change in relative abundance over time in other subjects. Community shifts over time in non-frozen samples are often associated with reductions in community diversity as profiles become dominated by a limited number of taxa (*Choo, Leong & Rogers, 2015*) due to factors such as altered temperatures relative to the normal human body or exposure to oxygen favouring aerobes and/or facultative anaerobes (*Chu et al., 2017*). However, subject B's profiles were increased in diversity (Fig. S3) due to depletion of a dominant *Prevotella* ESV from an average 26.0% relative abundance to 11.4% and 14.6% under ambient and chilled temperatures, respectively (Table S5). These results indicate that how stool communities respond to non-frozen storage varies by individual due to their specific configurations in stool microbial communities, and generalisations as to how stool community profiles respond over time after collection have to be made with caution. As *Prevotella* constitutes a major component of human gut communities and is a common feature in a large proportion of the human population (enterotype 2 described in *Arumugam et al., 2011*), changes as seen in subject B's profiles could severely impact the outcome of human population-based microbial community surveys.

The various human stool microbial community profiling studies mentioned here, like most other gut microbiome studies, involve homogenising samples to potentially reduce variability associated with subsampling (*Vandeputte et al., 2017*; *Sinha et al., 2016*; *Choo, Leong & Rogers, 2015*; *Tedjo et al., 2015*; *Carroll et al., 2012*). The issue with subsampling small portions of a stool for DNA isolation is that microenvironments can harbour specific taxa as shown by a quantitative PCR study measuring abundances of common human gut taxa in non-homogenised stools (*Gorzelak et al., 2015*). Using 16S-based community profiling, we also detected significant differences in overall community composition in two of three subjects when subsamples derived from different fractions of the same stool were compared. Therefore, we recommend based on these observations collection of a larger stool sample and inclusion of a homogenisation step before samples are stored frozen. That said, differences in community composition linked to within-sample variability is relatively minor compared to other experimental factors such as inter-individual and temporal

variation (*Voigt et al., 2015*) and is unlikely to influence interpretation of a multi-subject stool community survey.

The biggest limitation in this study is the limited number of subjects involved as we subsequently showed that inter-individual variability largely influences how profiles respond to the storage conditions tested. Future surveys should include a larger population of subjects to capture greater variability in stool community composition, which may help discern general patterns in how stool taxa respond to storage conditions. Another issue not addressed here is the influence of oxygen on stool community composition. One study demonstrated that exposure to oxygen primarily lowered relative abundances of *Faecalibacterium* and *Megamonas*, while increasing that of *Bacteroides* (*Chu et al., 2017*). These alterations could be minimised by storing samples in air-tight containers with oxygen scavengers, but we have not incorporated such elements in our sample collection workflow as it introduces additional inconveniences to participants of stool community surveys during self-sampling. Furthermore, Chu and colleagues showed that using standard 16S sequencing, community profiles from stools processed under anaerobic and aerobic conditions equally varied from control samples (anaerobic with cysteine as reducing agent to remove oxygen from solution) (Fig. 1A in *Chu et al., 2017*). Lastly, when comparing the findings presented here with other studies attention should be paid to the primers used as primer choice is widely known to influence 16S-based community compositional profiles (*Walker et al., 2015*; *Engelbrektson et al., 2010*).

## CONCLUSION

Stool sample delivery conditions and subsampling biases are known to alter microbial community composition. The use of preservative media, while meant to lessen such alterations, resulted in non-systematic, non-taxa specific but subject-specific changes in stool community profiles. However, when compared to inter-individual differences and community variations within an individual over time, effects due to delivery conditions and biases in sample fractions are small and unlikely to obscure inter-individual variability in surveys involving multiple subjects. Nevertheless, the effects of sample collection strategies should be given consideration especially when dealing with sensitive applications such as faecal microbiota transplants, which are reliant on exact resident microbial species.

## ACKNOWLEDGEMENTS

We wish to thank the stool donors for participating in this study, and reviewers for providing excellent feedback and suggestions.

### Funding

This study was supported by a seed fund for gut microbiota research provided by the Faculty of Medicine, The Chinese University of Hong Kong. The funders had no role

in study design, data collection and analysis, decision to publish, or preparation of the manuscript.

## Grant Disclosures
The following grant information was disclosed by the authors:
Faculty of Medicine, The Chinese University of Hong Kong.

## Competing Interests
The authors declare there are no competing interests.

## Author Contributions
- Yun Kit Yeoh conceived and designed the experiments, analyzed the data, prepared figures and/or tables, authored or reviewed drafts of the paper, approved the final draft.
- Zigui Chen analyzed the data, prepared figures and/or tables, authored or reviewed drafts of the paper, approved the final draft.
- Mamie Hui analyzed the data, approved the final draft.
- Martin C.S. Wong, Wendy C.S. Ho and Miu Ling Chin performed the experiments, contributed reagents/materials/analysis tools, approved the final draft.
- Siew C. Ng and Francis K.L. Chan conceived and designed the experiments, authored or reviewed drafts of the paper, approved the final draft.
- Paul K.S. Chan conceived and designed the experiments, analyzed the data, contributed reagents/materials/analysis tools, authored or reviewed drafts of the paper, approved the final draft.

## Human Ethics
The following information was supplied relating to ethical approvals (i.e., approving body and any reference numbers):

This study has been approved by the Joint Chinese University of Hong Kong-New Territories East Cluster Clinical Research Ethics Committee (reference number 2016.707).

## DNA Deposition
The following information was supplied regarding the deposition of DNA sequences:

Raw sequence data generated for this study are available in the Sequence Read Archive under BioProject accession PRJNA450690.

## Data Availability
Raw data is provided in the Supplemental Files.

## Supplemental Information
Supplemental information for this article can be found online at http://dx.doi.org/10.7717/peerj.6172#supplemental-information.

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
