# Peer review of "Impact of inter- and intra-individual variation, sample storage and sampling fraction on human stool microbial community profiles"

_PeerJ, doi:10.7717/peerj.6172_

## Round 0.1 · original submission · Minor Revisions

Your article has been reviewed by three referees who have all highlighted its relevance and timeliness. I am happy to inform you that only minor revisions are required. All three referees have provided detailed feedback on your article, which should be addressed during revision. In particular, please ensure all statistical tests and analyses are reported in the text, as it has been noted this information is missing in some instances.

I look forward to receiving your revised article in due course.

·

Basic reporting

This article by Yeoh and colleagues describes a pilot study (n = 3) to assess the optimal form of stool sample collection for a future population-based cohort study. The article is very well-written and easy to follow. The tables and figures are mostly easy to follow (comments on this below) and the sequencing data used is readily available. Ethics and consent were treated adequately.

Experimental design

The work presents three main weaknesses, two of which are discussed by the authors, namely the small number of subjects involved and the choice of only one preservative for the stool samples. Neither of these weaknesses invalidates the findings presented.

The third weakness, which is not mentioned by the authors, is the very shallow sequencing achieved. In my experience, at least 10,000 reads are necessary to provide a good overview of a stool sample (see, for instance, figure 6 in the pre-print https://doi.org/10.1101/286526). Thus, it is possible that some of the variability observed between different biological replicates is due to stochastic effects of OTU being included or not in a given sample. One way to assess this would be to perform a few random subsamplings from each dataset and assess whether the resulting differences are smaller or comparable to the differences between biological replicates observed.

Validity of the findings

It could be that Subject B does not have a particularly higher diversity or a different pattern of temporal changes in their stool samples, but rather that this Subject’s samples have been sequenced systematically deeper than the other two. A simple way to discard this would be to include a boxplot of sampling depth per subject in figure S1.

Additional comments

While the authors correctly discuss the effect of temperature and overgrowth of opportunistic organisms, I believe the discussion could be improved by considering the role of exposure to oxygen and thus selective killing of strictly anaerobic organisms. Is there any viable way of excluding oxygen from the samples that the authors have considered or could consider? Has this been approached by the existing literature?

Besides the issues discussed above, there are minor issues in the reporting of results that can easily be fixed, namely:
- The centroid sequence of each OTU is not provided. This can be done as an additional column in Suppl File 1 or as an additional fasta file
- The OTU ID in Suppl. Table 1 and 5 doesn’t match the ID provided in Suppl. File 1
- The legend in figure S1 refers to “species”. It would be more appropriate to use OTU or ASV (amplified sequence variant)
- In table S3, please inform the OTU ID, not only the taxonomic annotation
- Figures S2-S4 came with a large portion of grey background around them, probably an artefact from Inkscape
- I couldn’t understand what the different colours used in the text in figure S2 are supposed to mean
- In figure S3, it would be useful to see which of the presented differences are statistically significant

·

Basic reporting

The article is well structured and well written. The authors use clear and unambiguous, professional English throughout the article. However, in some occasions sentences are unnecessary long and could be simplified, such as in lines 207-210, 315-318, or typing errors occurred, line 240.

In general literature references are up to date and sufficient field background/context is provided. The section on reduction of community diversity in the case of highly abundant taxa in the discussion (lines 301-303) would benefit from a brief explanation of the mechanisms at play. In addition, the finding on dominant Prevotella abundances being severely reduced with storage conditions, is a major one in the light of the importance of this taxon in the determination of gut community types. Referencing to papers describing these gut community types (or enterotypes) would demonstrate how the work fits into the broader field of knowledge. Next to temperature, the possible role of (an)aerobic conditions should also be discussed as diverging factor in the discussion on specific taxa abundance changes.

The authors provide most of the necessary information throughout the text and in figures and tables. All appropriate raw data has been made available in a public database.
At some occasions, information on the statistical tests and its results are missing, such as in lines 180-181 and 283.
Figures and tables are of sufficient resolution, but description and labeling can be improved. Suggested improvements:
• Figure 1. Reduction in text and better use of symbols to make the study set up easier to understand at a glance. I have attached an example.
• Figure 2. Place data of Subject A in panel A, B in B, C in C and overall in D. Take legend of PCA overall out of the plot and move it to the side, as it applies to all of the plots. Number all sample points according to the day they were taken, remove text-box. Add info on storage duration (2,5,24h) in individual plots. Add info on the exclusion of samples (failed sequencing, …) in the figure legend – referring to the methods.
• Table 1: Heading could be turned into ‘all samples’. Sampling occasion might be better formulated as sampling time point. It could be useful to include which kind of levels the parameters include within the table of table legend. (e.g. for sampling occasion: is it all days, or period 1/2/3? Especially to compare with subject:sampling occasion). At this moment it seems like the column names of the data tables used during analysis by the researchers - who are fully accustomed with the research design – are used. Parameters should be turned into easy understandable text and convey all necessary information in an easy way for the reader.
• Table 2: Idem to suggestions Table 1.

Experimental design

This paper tries to assess the impact of sample storage conditions on stool community profiles of Asian individuals. The research question is well defined, relevant & meaningful. The authors state in the abstract and introduction how their research fills an identified knowledge gap. They performed a rigorous investigation, complying with the ethical standard and using the latest available methods (e.g. application of the Dada2 algorithm for sequence variance identification).

The methods are well described and provide enough detail to repeat the analysis. I would suggest to also include information on the number of unassigned taxa at each taxonomical level at the end of the paragraph ‘16S sequencing data processing’. The authors chose to apply a centered log-ratio transformation for the PCA and state this in the result section, yet I think it is more appropriate to include this in the method section, together with a brief explanation on the reasons to do so.

Validity of the findings

The authors used the appropriate materials and methods and performed appropriate statistical tests, including multiple testing correction where necessary. Some notes:
• Lines 180-182, statements are not completely supported by the results (I think one cannot state that the specific durations of 2,5 and 24 hours are associated with shifts in community composition but only duration in general).
• Lines 230-232: A much higher amount of taxa is enriched versus depleted over time, could the authors elaborate on this? This might be a consequence of the relative data.
Importantly, the data on which the conclusions are based are made available for the community.

Additional comments

This work contains some important messages for our field. Thank you for taking the time to look into this subject and report the results of your analysis in a clear way. In order to improve the manuscript further, I would like to make the following suggestions.
• Include a ranking of the variables according to their effect on microbiota composition.
• Make optimal use of the longitudinal set up of the study by determining not only the inter-individual variation but also intra individual variation. For instance, using Bray Curtis dissimilarity or another distance metric, variation between samples could be determined between persons (inter-ind.), between time-points of the same time-period (intra ind.), storage methods (for the same person, at the same time point), intra-sample.
• For the title, ‘delivery conditions’ does not really convey what it should. With the numerous reports on the microbiome of infants, the wrong association with childbirth is easily made. ‘Storage conditions’ might be more appropriate. In order to make the title and text more comprehensible a distinction between biological relevant variation (inter-ind, temporal, intra-sample) and induced variation by storage conditions could be made.
• To be completely correct, one should make a distinction between the taxonomic units obtained using the Dada2 workflow and those obtained by previous methods (Operational Taxonomic Units (OTUs), defining taxonomic units based on a cut off value for sequence similarity). With dada2 the taxonomic units are of a higher resolution and can be named amplicon sequence variants (ASVs).

Reviewer 3 ·

Basic reporting

No comment

Experimental design

No comment

Validity of the findings

No comment

Additional comments

The manuscript under review describes work that is relevant and current. Although the work in part represents replication of the previously reported data the rationale for such approach is explained and is convincing. Special value of the work rests in the meticulous description of the sequencing data analysis as well as the appropriate and valid use of statistical approaches. On the other side, the low initial number of stool donors (n=3) is a statistical power issue which is somewhat compensated with multiple donor sampling timepoints.
There are only several points on which the manuscript should be improved:
1. There should be original reference added for primers used in 16S amplicon sequencing: Klindworth et al. Nucleic acid research 2013;41(1):e1
2. In the methodology section describing the amplicon sequencing there should be information added that describes the controls used for verifying the quality of the sequencing runs (positive, negative).
3. In the discussion section more attention should be given to previous research performed with other preservation buffers (RNA later, OMNIgene gut sysem etc) so to provide broader context of the presented work
4. The sentence: ”In addition, the findings presented here should also be verified using different sets of universal 16S primers as primer choice is widely known to influence 16S-based community compositional profiles” although in its essence true, should be rephrased in a way to reflect the fact that 16S primers influence 16S-based community compositional profiles but not to encourage other researchers to use different primers. Different studies performed in different laboratories and with only slight variations in the methodology produce sufficiently different results disabling appropriate study comparisons. Studies could only be compared if identical protocols/methodologies are used and the future of any long lasting technology/methodology lies in standardized methodology. The suggested wording would potentially be: “In addition, when comparing the findings presented here with other studies attention should be paid to the primers used for sequencing as primer choice is widely known to influence 16S-based community compositional profiles”.

---

## Round 0.2 · Minor Revisions

Thank you for making the requested amendments to your article. There are a still a couple of very minor editorial issues to be dealt with, as noted by one reviewer.

- Tables use the term OTU, while the text uses ESV throughout. Please change OTU to ESV in Tables.

- In the discussion, please use "microbiota, "microbiome", "microbial community" or "bacterial community" instead of "flora".

I will accept your article as soon as a new version with the minor changes incorporated is submitted to PeerJ.

·

Basic reporting

no comment

Experimental design

no comment

Validity of the findings

no comment

Additional comments

All my previous concerns have been addressed. There are only very minor issues left, namely:
- Tables use the term OTU, while the text uses ESV throughout
- In the discussion, the term "stool flora" is inadequate in a professional setting. While perfectly acceptable in outreach activities, "flora" in a discussion between biologists refers to Viridiplantae. "Microbiome", "microbial community" or, even more precisely, "bacterial community" should be preferred.

·

Basic reporting

No comments.

Experimental design

No comments.

Validity of the findings

No comments.

Additional comments

The authors sufficiently addressed all outstanding issues in this version. I have no further comments and support the publication of this work in PeerJ.

Reviewer 3 ·

Basic reporting

No comment

Experimental design

No comment

Validity of the findings

No comment

Additional comments

It is a pitty that no positive controls of sequencing run were included, this would additionally add value and context to the reserach performed and results obtained.

---

## Round 0.3 · accepted · Accept

Thank you for making the requested minor changes to your article. I look forward to seeing it in print.

#